# Strain Rate and Porosity Effect on Mechanical Characteristics and Depolarization of Porous Poled PZT95/5 Ceramics

**DOI:** 10.3390/ma13214730

**Published:** 2020-10-23

**Authors:** Zhaoxiu Jiang, Guangfa Gao, Xiaofeng Wang, Yonggang Wang

**Affiliations:** 1School of Mechanical Engineering, Nanjing University of Science and Technology, Nanjing 210094, China; jiangzhaoxiu@njust.edu.cn (Z.J.); gfgao@ustc.edu.cn (G.G.); 2Key Laboratory of Impact and Safety Engineering, Ministry of Education, School of Mechanical Engineering and Mechanics, Ningbo University, Ningbo 315211, China; wangxiaofeng@nbu.edu.cn

**Keywords:** porous PZT95/5 ceramic, uniaxial stress, depolarization, strain rate, phase transformation

## Abstract

Shock wave compression of poled PZT95/5 ceramics results in rapid depoling and a release of bound charge. Porous PZT95/5 ceramics are superior to dense ceramics in high-voltage breakdown resistance under shock-wave loading. In this article, the mechanical and electrical responses of porous poled PZT95/5 ceramics under uniaxial stresses at different strain rates were investigated using the servo-hydraulic MTS810 universal test machine and the improved split Hopkinson pressure bar system. The engineering stress vs. axial and radial engineering strain curves of porous poled PZT95/5 ceramics under different strain rates exhibit anomalous nonlinear behavior. The nonlinear behavior and depolarization mechanism of porous poled PZT95/5 were attributed to the domain switching and phase transformation. By comparing the stress–strain curves of the porosity porous poled PZT95/5 ceramics at different strain rates, an obvious strain rate sensitivity of mechanical behavior can be found, and the strain rate sensitivity decreases with the increase of porosity. The critical stress of domain switching and phase transformation and the strength increased with increasing strain rate. In addition, their normalized values showed a logarithmic relationship with the strain rate. Finally, we suggest that the maximum polarization released is nearly independent of stress state and strain rate, and it only depends on the porosity.

## 1. Introduction

Zirconium-rich lead titanate Pb(Zr_0.95_Ti_0.05_)O_3_(PZT95/5) ferroelectric ceramics exhibit a transformation from the ferroelectric phase (FE) to the antiferroelectric phase (AFE) in response to compressive stress [1,2,3,4], and they have been utilized in pulsed power applications for many years [4,5,6,7]. In these applications, electrical energy is stored in the PZT95/5 ceramics by an initial poling process and released into an electrical load by an externally applied shock-wave loading. It has been demonstrated that mechanical failure may often cause dielectric breakdown in ceramics, and the porous ceramics are superior to dense ceramics for preventing high-voltage breakdowns during shock-wave loading [8,9,10].

Based on pore structure advantage, the pore structure of the electrical and mechanical properties of porous PZT95/5 ceramics has been extensively studied in recent years. Zeng et al. [11] explained that the porous PZT95/5 ceramics with spherical pores exhibited better properties than irregular and lower dielectric loss and transformation pressure than dense PZT ceramics. Setchell [12] and Feng et al. [13] conducted uniaxial–strain experiments on porous poled PZT95/5 ceramics. Their results indicated that the mechanical and electrical shock properties of porous PZT ceramics were insensitive to microstructural differences in materials of the same density, and the charge-releasing rate was faster than the dense samples. Moreover, as part of an ongoing effort to understand the detailed nonlinear mechanical behavior of PZT95/5 ceramics, we have carried out a large number of quasi-static uniaxial compression tests [14]. We showed that the nonlinear deformation process was caused by domain switching (DS) and phase transformation (PT), which was proved by analyzing the axial strain vs. radial strain curves. On the other hand, the experiments on various ceramics have demonstrated that the failure of brittle material has obvious strain rate sensitivity and it strongly dependent on defects (i.e., voids, cracks) [15,16,17]. However, although most works were almost exclusively concerned with the behavior of porous PZT95/5 ceramics under shock-wave loading with strain rates from 10^4^ s^−1^ to 10^5^ s^−1^ and quasi-static loading with strain rates from 10^−4^ s^−1^ to 10^−1^ s^−1^, there is still a lack of systematic study about the mechanical response and depolarization characteristics of porous PZT95/5 ceramics at intermediate strain rates (10^2^ s^−1^–10^3^ s^−1^) and the strain rate effect on it. Thus, it is necessary to study the effect of strain rate on depolarization and nonlinear mechanical characteristics of porous PZT95/5 ceramics further, whether examining porous PZT ceramics’ nonlinear characteristics as a scientific issue or expanding its application range under different loads.

In this paper, the poled PZT95/5 ceramics as a function of systematic changes in porosity were prepared by sintering compacts consisting of PZT and pore formers. A series of uniaxial stress experiments of porous ceramics were carried out using an improved split-Hopkinson pressure bar (SHPB) system and the servo-hydraulic MTS810 universal test machine. The effects of strain rate on the mechanical characteristics and depolarization behavior will be discussed.

## 2. Experimental

### 2.1. Sample Preparation

Four kinds of PZT 95/5 ceramics with different porosities were prepared by mixing PZT powders with PMMA (polymethyl methacrylate). The pore former PMMA was spherical 30 μm in diameter, and 0 wt %, 1.0 wt %, 2.0 wt %, or 3.0 wt % PMMA were added into the synthesized PZT powders. The mixed powders were firstly pressed at about 200 MPa to a Φ9 mm and Φ14 mm column with a height of 8 mm; then, they were processed into columns with Φ6 mm × 6 mm and Φ12 mm × 6 mm for testing at a high strain rate and low strain rate, respectively. At last, samples were coated with silver electrodes and poled in a silicone bath at 120 °C for 10 min, and the polarization electric field was 3000 V/mm. Figure 1 shows the SEM images of porous PZT95/5 ceramics with different porosity, which were collected on the samples’ cross-sectional cutting surface. The bulk density was measured using the Archimedes method, and the porosity of the sample was calculated from the ratio of the bulk density to theoretical density (8.08 g/cm^3^) [1]. The measured results are shown in Figure 1 and Table 1.

### 2.2. Testing Method

Uniaxial compressive tests of porous poled PZT95/5 ceramics at a low strain rate were performed on a servo-hydraulic MTS 810 universal test machine. The implementation method can be referred to in our previous study [14]. The intermediate strain rate uniaxial compressive tests were performed using an improved split Hopkinson pressure bar (SHPB for short) system, as sketched in Figure 2. The whole set-up mainly consists of a traditional SHPB device and a DIC (Digital Image Correlation) measurement system [17,18,19,20,21]. All bars and strikers are made of high-yield steel and have the same diameter of 14.5 mm. In traditional SHPB testing, based on the one-dimensional elastic stress wave theory, the engineering strain rate $\left(\dot{\varepsilon }\right)$, engineering strain (ε,eng.strian), and engineering stress (σ, eng.stress) of the sample can be calculated under the assumption of one-dimensional stress loading and uniform distribution of stress and strain along the sample length, and the estimation formula is as follows:(1)$$\sigma \left(t\right)=\frac{A_{b}E_{b}}{A_{s}}\varepsilon_{t}\left(t\right)$$
(2)$$\dot{\varepsilon }\left(t\right)=-\frac{2c_{b}}{L_{s}}\varepsilon_{r}\left(t\right)$$
(3)$$\varepsilon \left(t\right)=\int -\frac{2c_{b}}{L_{s}}\varepsilon_{r}\left(t\right)dt$$
where $E_{b}$, $c_{b}$, and $A_{b}$ are the elastic modulus, longitudinal wave speed, and cross-sectional area of the bar. $A_{s}$ is the cross-sectional area of the sample. Furthermore, $\varepsilon_{i}\left(t\right)$, $\varepsilon_{t}\left(t\right)$, and $\varepsilon_{r}\left(t\right)$ are the strains of the incident, transmitted waves, and reflected waves, respectively. These strains are measured utilizing strain gauges bonded in the incident and transmitted bar.

As mentioned above, using Equation (3), strain values averaged over the length of the sample are obtained. However, for the brittle material, it is difficult to achieve stress/strain equilibrium and constant strain rate loading creates an ineffective testing time because of its small failure strain. There are many discussions on the stress uniformity of samples. Most of the methods involve placing a pulse shaper at the front of the incident bar. The striker can impact the pulse shaper first and prolong the loading time to achieve the quasi-static stress equilibrium and constant strain rate before the sample is destroyed. Figure 3 shows the original voltage signals of the incident wave, reflected wave, and transmitted wave recorded in SHPB tests. As shown in the figure, before the sample is destroyed, the obvious plateau line appears in the reflected wave curve, which indicates that the constant strain rate loading is realized by using the pulse shaper technique [22,23,24,25]. On the other hand, for the uniformity of stress/strain distribution, besides the stress equilibrium during loading, whether the local stress concentration may still lead to the strain uniformity is not well satisfied [22]. To study the effect of stress concentration on the uniformity of strain distribution, lubricant was used in the experiment to avoid the influence of friction [26], and a DIC measurement system with an ultra-high-speed camera and two high powerful flashlights was used to obtain the full-field strain of the sample, as shown in Figure 2. The review of the DIC method was presented by Sutton et al. [18]. An ultra-high-speed camera records the sample’s deformation with 1 million fps (frames per second) (the maximum frame rate is 5 million fps, the image resolution is 924 × 768 pixels). Before testing, a speckle pattern is applied to the sample and its sides with spray paint, as shown in Figure 2b. The deformation sample-recorded images during experiments were subsequently analyzed by the commercial VIC-2D software (Vic-2D 2, Correlated Solutions, Inc., lrmo, SC, USA) to determine the field strain.

In addition, the compressive load direction is parallel to the poling direction of poled ceramics. A 92 ohm resistor (R) is connected in parallel with the ends of the silver-plated electrode for the sample, as shown in Figure 2. The electrical characteristics of poled ceramics are monitored by measuring the voltage at both ends of the resistor, as is shown in Figure 3. According to the principle of one-dimensional bar elastic wave propagating, we can move the transmitted wave to the middle of the incident wave’s starting times and the reflected wave’s starting times. The stress and discharge voltage are synchronized in time by shifting the transmitted wave, as is shown in Figure 4. The following equation can calculate the remnant polarization of porous poled ceramics in the uniaxial compression experiment:(4)$$P_{r}=\frac{R}{A}\int U(t)dt$$
where *P_r_* is the remnant polarization, *U* is the output voltage from the experiment, and *R* is a resistor of 92 ohm.

## 3. Results and Discussion

### 3.1. Stress–Strain Curves and Depolarization Characteristics at the Intermediate Strain Rate

Figure 5 shows the time evolution of the axial strain distribution (the corresponding strain values are determined by the DIC method) during the SHPB testing and the contours of strain at 68 μs. As shown in Figure 5, the strain distribution along the sample length is non-uniform except in the middle region. Such non-uniformity of strain may have resulted from the stress concentration near the contact interface [18]. To correct the effect of stress concentration on the traditional SHPB strain measurement and verify the DIC method, a strain gauge (size ≈3 mm) was placed in the middle of the sample. Figure 6 shows the history curves for eng.strain obtained by traditional SHPB, strain gauge, and the DIC method, respectively. Among them, the strain measured by the DIC method is the uniformly distributed average strain of the sample. It can be seen that the strains measured by the DIC method and strain gauge agree well with each other, which verifies the validity of the DIC method in this experiment. However, the strain of traditional SHPB measured is bigger than that of the above two results. Therefore, we selected the uniformly distributed average strain of the DIC method measured as the axial eng.strain and form the eng.stress–eng.strain curves. As shown in Figure 7, the eng.stress–eng.strain curves of poled ceramics with different porosity at the strain rate of ≈300 s^−1^ are obtained by the improved SHPB experiment.

Take porous poled PZT95/5 ceramics with ≈11% porosity for example; the nonlinear characteristics of porous poled ceramics will be analyzed. Representative plots of axial eng.stress versus radial eng.strain and axial eng.stress versus axial eng.strain for porous poled ceramics with ≈11% porosities at the strain rate ≈300 s^−1^ are shown in Figure 8. Initially, the axial strain and radial strain exhibit a linear behavior with increased stress. This linear behavior corresponds to the elastic response of the ferroelectric phase of pole ceramics. After exceeding the stress corresponding to point “A”, both the radial and axial strains depart sharply from the linear deformation. As the stress increases, the radial and axial strain deviate linearly after exceeding the stress corresponding to point “B”. The deformation in radial direction reverses from tensile to compressive at the stress corresponding to point “C”. As described above, the nonlinear behavior of porous poled ceramics at intermediate strain rate is similar to that of unpole PZT ceramics under quasi-static conditions [14]: (1) the linear slope of ceramics near failure is almost the same as that at the beginning of loading; (2) the anomalous temporary reversal also appears on the curve of radial eng.strain–axial eng.stress curve. According to the similarity of the nonlinear characteristics of stress–strain curves between porous poled ceramics at an intermediate strain rate and unpoled ceramics under quasi-static conditions, we can rule out the possibility of void collapse. Thus, the stress corresponding to points “A” and “B” in Figure 8 can be defined as the critical stress of DS ($\sigma_{D}$) and PT ($\sigma_{P}$), respectively [14]. After that, the linear behavior of axial strain versus stress curve occurs again, which corresponds to the linear elastic behavior of poled PZT95/5 ceramics in an AFE phase. The stress level of point “A” in Figure 8 can be defined as critical stress for the DS, which is denoted as $\sigma_{D}$.

As discussed above, the nonlinear behaviors of the poled ceramics under the intermediate strain rate are attributed to the DS and PT. For the poled ceramics, poling aligns as many dipoles as possible parallel to the axis of compression. When it was subjected to axial compression stress, the dipoles’ direction deflects and becomes perpendicular to the axis of compression [27]. It can release a large number of charges to form a circuit through field resistance, as shown in Figure 4. Figure 8 also shows the polarization released versus the axial stress curve for porous poled PZT95/5 ceramics under the intermediate strain rate. At the beginning of the loading process, a small charge is released due to the ferroelectric ceramics’ piezoelectric effect. Stress-induced depolarization occurs up to relatively high stress, and this stress level corresponds to $\sigma_{D}$. With increasing load, the stress-induced depolarization continues even after the beginning of PT and the end of it at the stress of point “D”. Therefore, the stress-induced depolarization of poled ceramics is attributed to the mechanisms of DS and PT.

### 3.2. Different Porosity Sample Tests at Low and Intermediate Strain Rates

The axial eng.stress–eng.strain curves at the strain rates of ≈10^−4^ s^−1^ and ≈300 s^−1^ of poled ceramics with four different porosities are shown in Figure 9a–d. As can be seen from Figure 9, the repeatability of the stress–strain curves for brittle materials with similar porosity under low and intermediate strain rates conditions is acceptable.

The compressive strength of poled ceramics versus porosity at the strain rate of 300 s^−1^ and 10^−4^ s^−1^ is shown in Figure 10. From a given range of the porosity in tests, the strength at the strain rate of 10^−4^ s^−1^ decreases linearly with the porosity of ≈5% to ≈15% and sharply drops at the porosity of ≈18% (the sharp strength drop may be related to the coalescence of pores, and the spacing between voids and between voids and defects), while the strength at the strain rate of 300 s^−1^ decreases nonlinearly with the increasing porosity. It is also worth noting that with the increase of porosity, the compressive strength of poled ceramics at the strain rate of 300 s^−1^ draws near the strain rate of 10^−4^ s^−1^. This indicates that the strain rate sensitivity decreases gradually with porosity.

According to the measured stress–strain curves of porous poled ceramics with different porosities in Figure 9, we also examined the porosity dependence of the critical stress for the initiation of DS and PT at the strain rate of 10^−4^ s^−1^ and 300 s^−1^, respectively. First, the critical stress of $\sigma_{D}$ and $\sigma_{P}$ are normalized as a function of porosity by $\sigma_{D0}$ and $\sigma_{P0}$, where $\sigma_{D0}$ and $\sigma_{P0}$ are the values of $\sigma_{D}$ and $\sigma_{P}$ for the materials at a porosity of 0%. These values were determined to be $\sigma_{D0}$ ≈ 108 MPa, $\sigma_{P0}$ ≈ 199 MPa at the strain rate of 10^−4^ s^−1^ and $\sigma_{D0}$ ≈ 156 MPa, $\sigma_{P0}$ ≈ 270 MPa at the 300 s^−1^ strain rate, which were obtained by linear fitting of the relationship between $\sigma_{D}$ and $\sigma_{P}$ and porosity. It is a pity that we have not found the anomalous temporary reversal of the radial eng.strain–eng.stress curves of the samples with ≈18% porosity. Therefore, the values of $\sigma_{P}$ for the samples with ≈18% porosity are not shown in Figure 11. The normalized $\sigma_{D}$ and $\sigma_{P}$ at the 10^−4^ s^−1^ and 300 s^−1^ strain rates show similar declining trends with the increasing porosity, as shown in Figure 11. Therefore, the relationship between the normalized $\sigma_{D}$, $\sigma_{P}$ and porosity could be expressed by a linear regression formula as follows:(5)$$\frac{\sigma_{D}}{\sigma_{D0}}=\frac{\sigma_{P}}{\sigma_{P0}}=1-cp$$
where *c* is a material-dependent parameter. Using the linear fit of normalized $\sigma_{D}$ and $\sigma_{P}$ versus porosity curves, parameter *c* is determined to be ≈0.026.

Finally, we examined the porosity dependence of the depolarization for poled ceramics at the low and intermediate strain rates. Figure 12 shows the polarization released of poled ceramics with different porosities recorded in servo-hydraulic MTS 810 universal and SHPB tests. The stress of initial depolarization and the peak polarization released decreased with increased porosity at the intermediate strain rate, and the stress of the initial depolarization is hardly affected by porosity at a low strain rate. Meanwhile, the peak polarization released of poled ceramics as a function of porosity shows a linear decrease from ≈4% to ≈15%. It is essentially consistent with hydrostatic compression tests [28], as is shown in Figure 13. For the sample with ≈18% porosity, the fracture occurred before the charge releases fully, so that the peak polarization released sharply decreased with the porosity under uniaxial stress.

### 3.3. Strain Rate Effect

The eng.stress–eng.strain curves of porous poled ceramics at approximately the same porosity (≈11%) with different strain rates are shown in Figure 14. The stress–strain curves of porous poled ceramics at different strain rates exhibit nonlinear characteristics for all of these tests. Compared with the quasi-static tests, poled ceramics’ compressive strength is significantly enhanced at the high strain rate. This significant increase in brittle materials’ strength has generally been explained as the strain rate effect [29,30]. The DIF (dynamic increase factor) is usually used to characterize brittle materials’ strain rate sensitivity, which is defined by the strength ratio under high strain rate to the strength under quasi-static tests [17]. Figure 15 shows the DIF of porous poled ceramics with ≈11% porosity as a function of strain rate. A sharp increase in strength was observed between quasi-static and dynamic loading conditions, as shown in Figure 15. The strain rate effect of porous poled ceramics can be expressed by the DIF logarithmic regression equation as follows:(6)$$DIF=1.63+0.096log\left(\dot{\varepsilon }+0.0014\right)$$

As discussed earlier, the nonlinear mechanism of porous poled ceramics under uniaxial stress loading is caused by DS and PT. For porous poled ceramics, the strain rate effect affects the compressive strength and the nonlinear mechanical characteristics, as is shown in Figure 16. The value of $\sigma_{D}$ at the strain rate of 300 s^−1^ is ≈98.8 MPa, which is much higher than the value of $\sigma_{D}$ (≈48.2 MPa) obtained for poled ceramics under quasi-static test. In addition, the value of $\sigma_{P}$ is ≈250.7 MPa at the high strain rate, which is higher than quasi-static conditions (≈117.2 MPa). Thus, the value of $\sigma_{D}$ and $\sigma_{P}$ show a very similar declining trend with the increase of strain rate, as shown in Figure 17. The correlation between normalized critical stress and strain rate is expressed by the formula similar to Equation (5):(7)$$\frac{\sigma_{D}}{\sigma_{Ds}}=\frac{\sigma_{P}}{\sigma_{Ps}}=0.78\log\left(\dot{\varepsilon }+134.84\right)-2.74$$
where $\sigma_{Ds}$ and $\sigma_{Ps}$ are the critical stress of DS and PT at the strain rate of 10^−4^ s^−1^. In addition, the stress-induced depolarization of porous poled ceramics is attributed to the mechanisms of DS and PT, so the correlation between the initial critical stress of depolarization and strain rate is also increased with the increasing strain rate, and the maximum polarization released is not affected by the strain rate.

## 4. Conclusions

The effect of strain rate on mechanical characteristics and poled ceramics’ depolarization were investigated under uniaxial stress. The nonlinear mechanical characteristics and stress-induced depolarization of poled ceramics at different strain rates are attributed to the mechanisms of DT and PS. The value of $\sigma_{D}$ and $\sigma_{P}$ and the strength increased with the increasing strain rate, and there was a very marked logarithmic relationship between the normalized value and strain rate. The stress-induced polarization released is not obviously related to strain rate and stress state, and it decreases with the increasing porosity. The strength under different strain rate conditions decreases with porosity, and the normalized critical stresses for DS and PT decreased linearly with increasing porosity.

## Figures and Tables

**Figure 1 materials-13-04730-f001:**
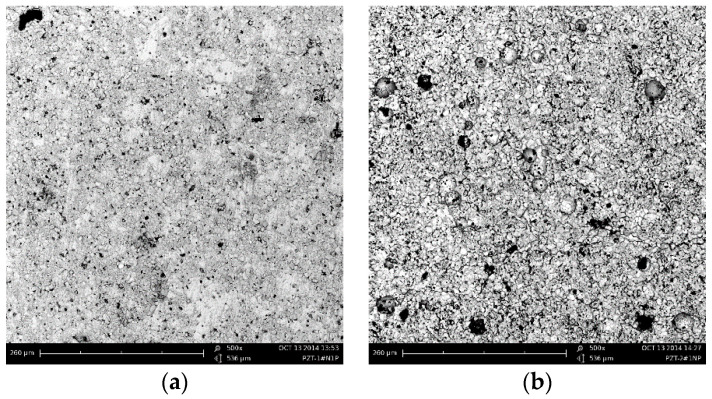
SEM micrographs of poled ceramics with different porosities: (**a**) porosity ≈5% (0 wt % PMMA); (**b**) porosity ≈11% (1 wt % PMMA); (**c**) porosity ≈15% (2 wt % PMMA); (**d**) porosity ≈18% (3 wt % PMMA).

**Figure 2 materials-13-04730-f002:**
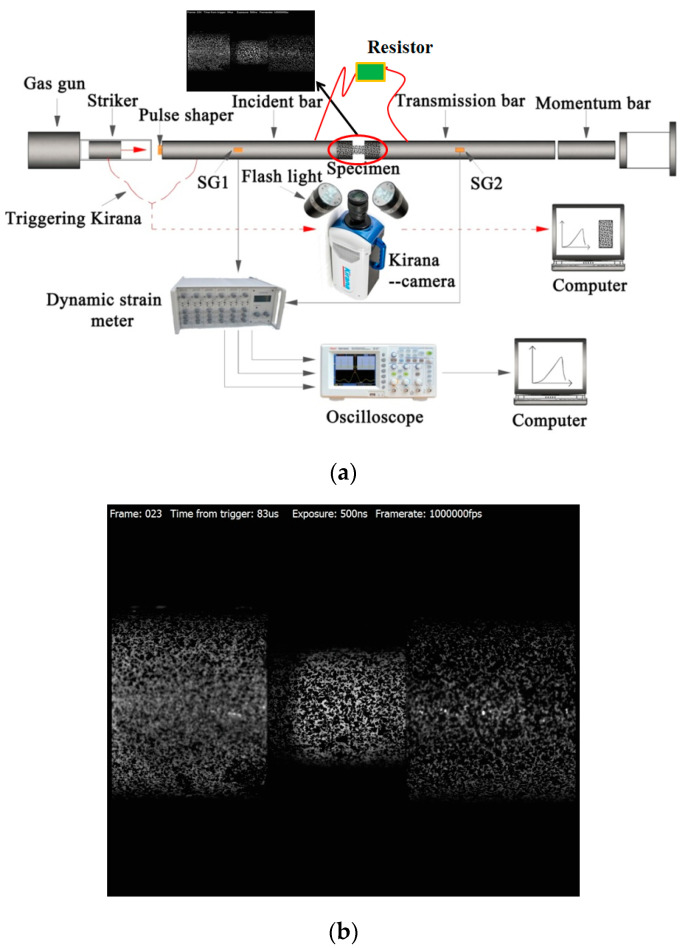
Test system: (**a**) Schematic of the split-Hopkinson pressure bar (SHPB) set-up and the DIC measurement system; (**b**) Surface speckle image of sample and bars.

**Figure 3 materials-13-04730-f003:**
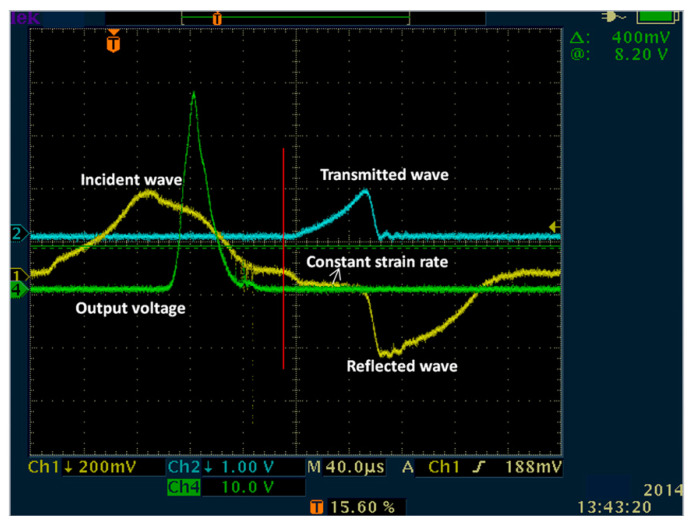
Date recorded in an SHPB test.

**Figure 4 materials-13-04730-f004:**
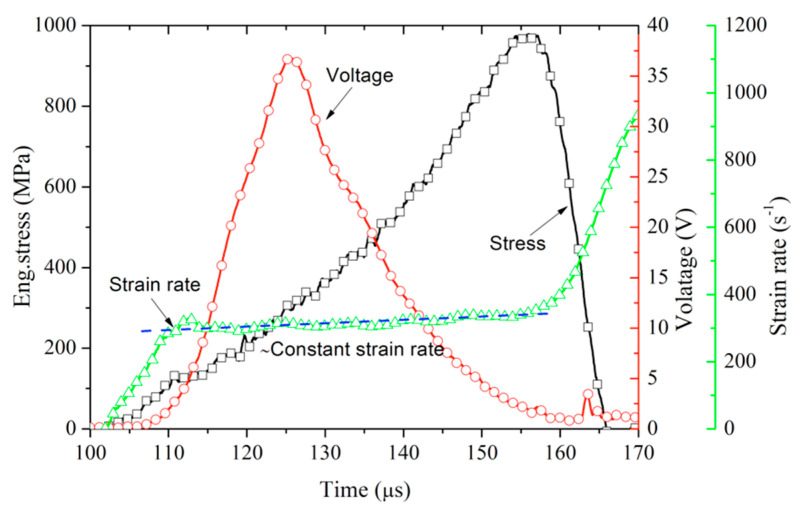
Profiles of strain rate, output voltage, and eng.stress for Pole PZT95/5 ceramic under high strain-rate loading.

**Figure 5 materials-13-04730-f005:**
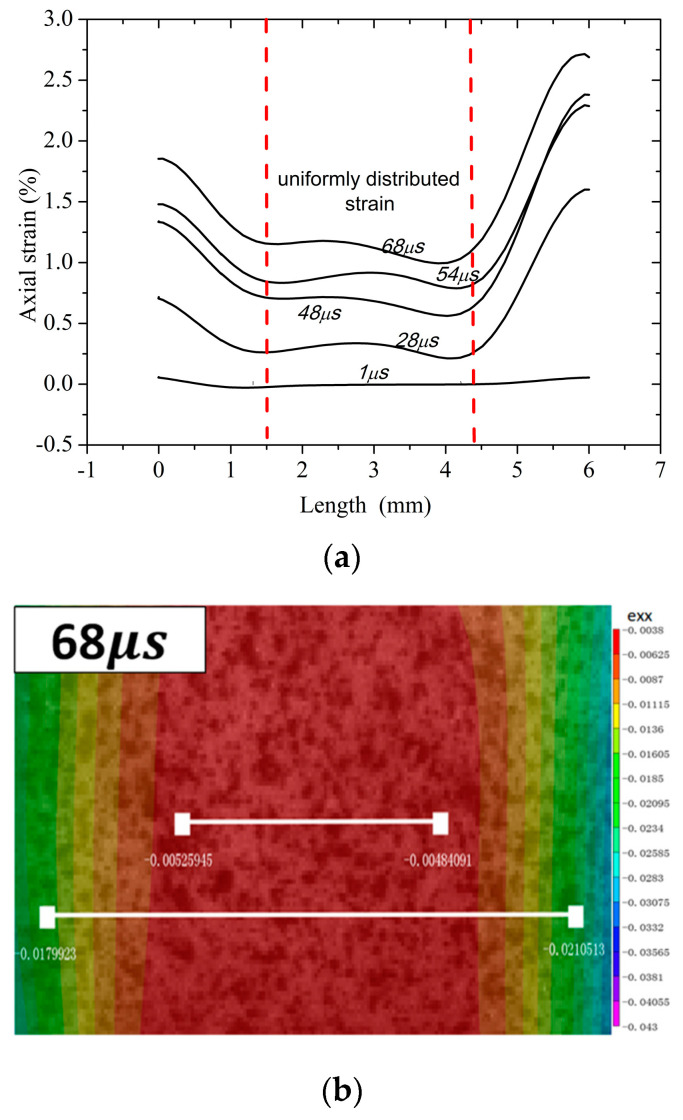
Axial strain distribution along the sample’s length: (**a**) different times; (**b**) the contours of strain at 68 μs.

**Figure 6 materials-13-04730-f006:**
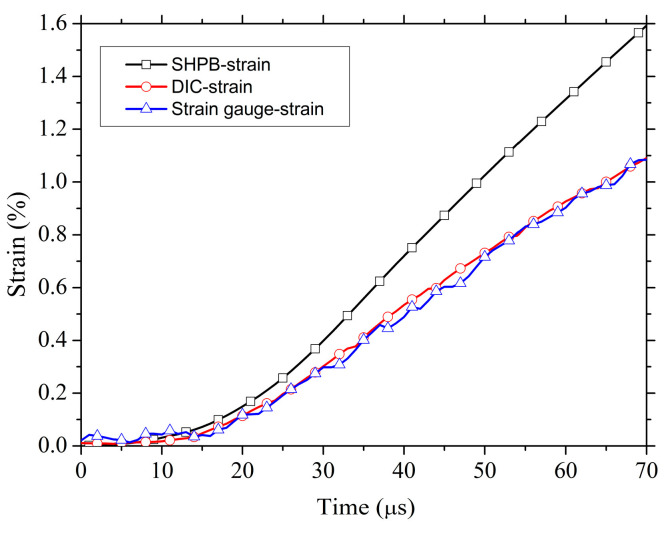
Comparison of time-dependent curves of different strain-measuring methods.

**Figure 7 materials-13-04730-f007:**
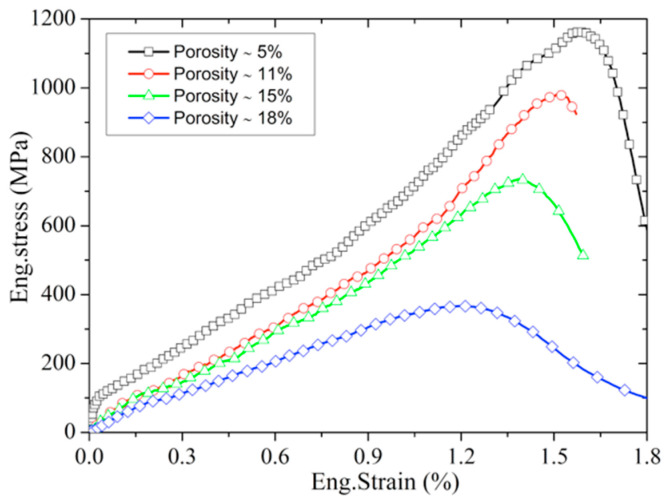
Typical eng.stress–eng.strain curves of poled PZT95/5 ceramics with different porosities at the strain rate of ≈300 s^−1^.

**Figure 8 materials-13-04730-f008:**
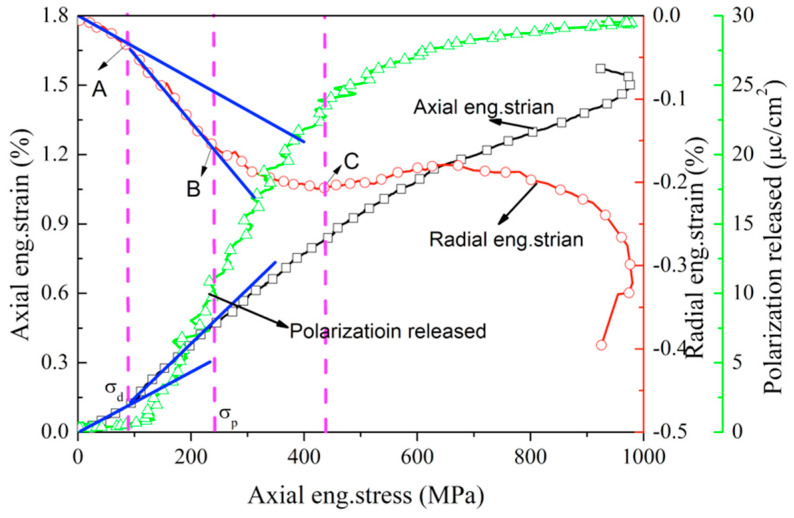
Axial eng.stress vs. axial eng.strain, radial eng.strain and polarization released curves for poled porous ceramics with ≈11% porosity at the strain rate of 300 s^−1^.

**Figure 9 materials-13-04730-f009:**
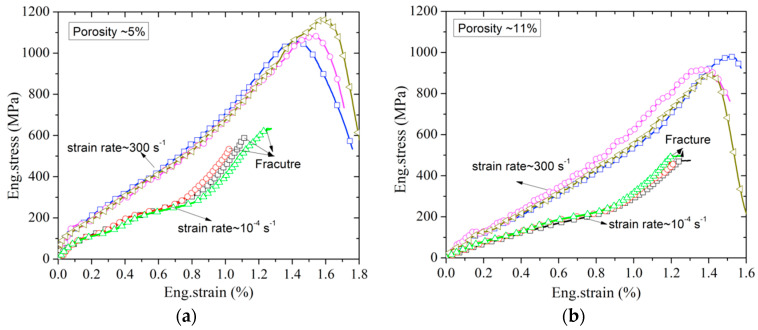
Eng.stress–eng.strain curves of poled PZT95/5 with different porosities at the strain rate of ≈300 s^−1^ and ≈10^−4^ s^−1^: (**a**) porosity ≈5%; (**b**) porosity ≈11%; (**c**) porosity ≈15%; (**d**) porosity ≈18%.

**Figure 10 materials-13-04730-f010:**
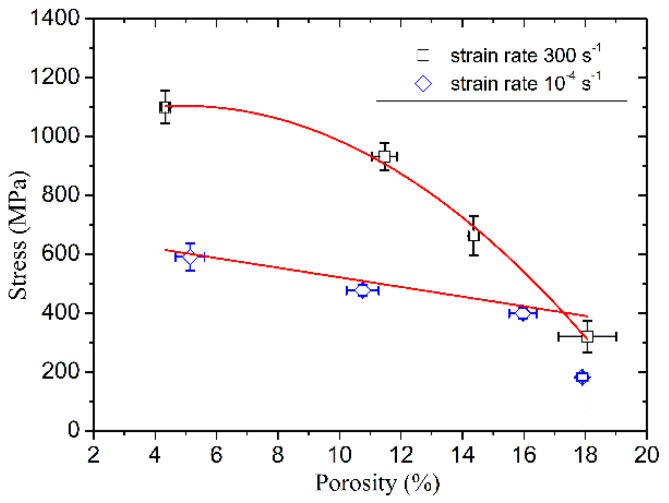
The compressive strength vs. porosity at the strain rate of 300 s^−1^ and 10^−4^ s^−1^.

**Figure 11 materials-13-04730-f011:**
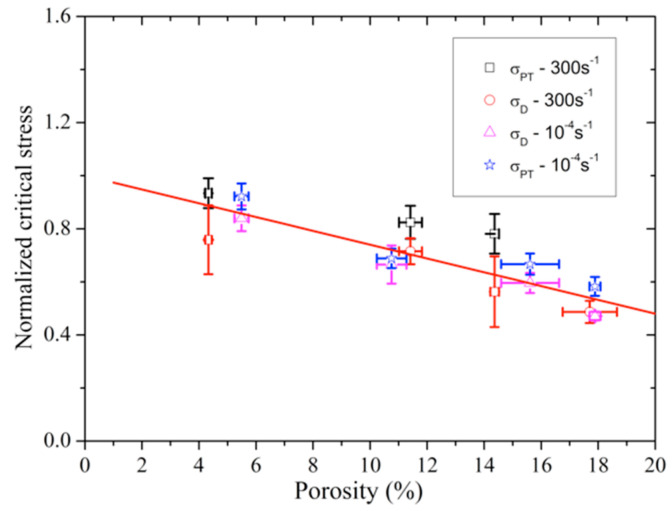
The normalized $\sigma_{D}$ and $\sigma_{P}$ versus porosity.

**Figure 12 materials-13-04730-f012:**
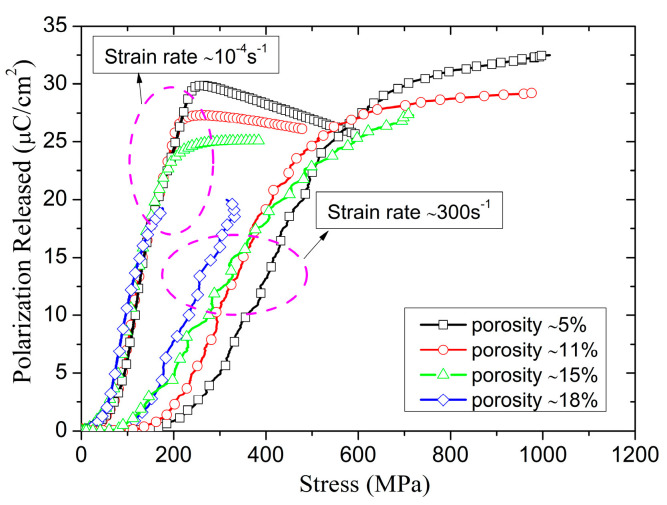
The polarization released of poled ceramics with different porosities under the strain rate of ≈10^−^^4^ s^−^^1^ and ≈300 s^−^^1^ as a function of stress.

**Figure 13 materials-13-04730-f013:**
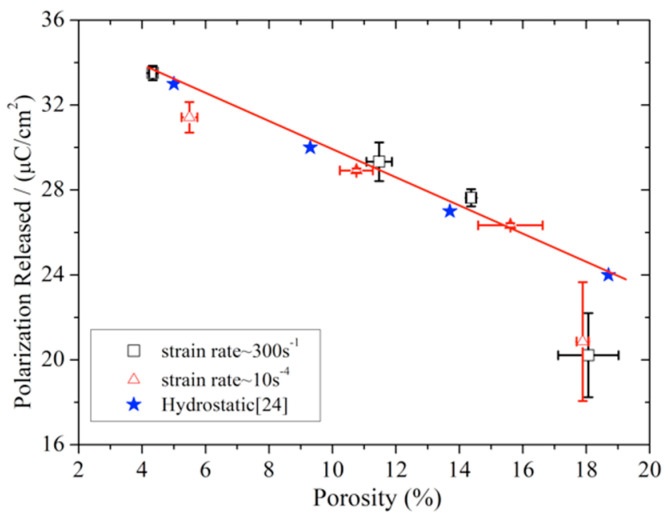
The peak of polarization released as a function of porosity for poled ceramics under different stress states.

**Figure 14 materials-13-04730-f014:**
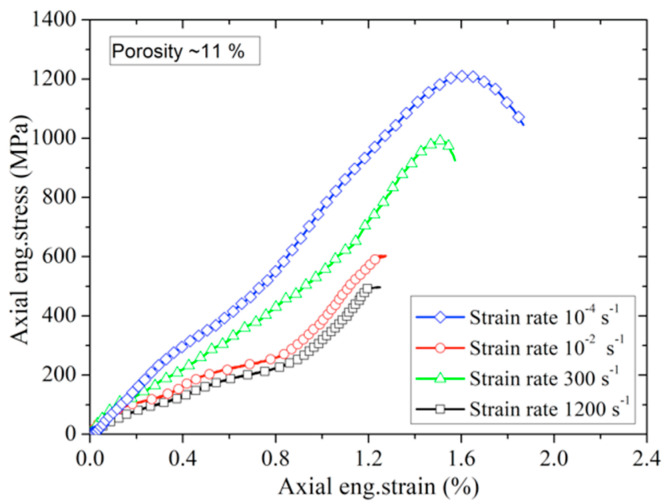
Eng.stress–eng.strain curves of porous poled ceramics at different strain rates.

**Figure 15 materials-13-04730-f015:**
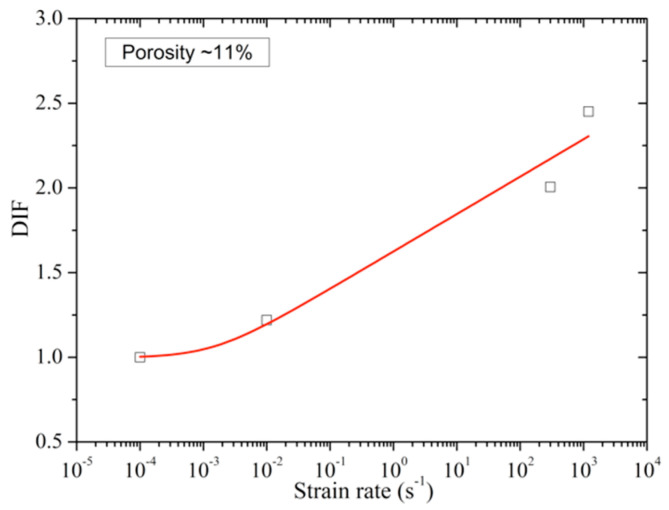
The dynamic increase factor as a function of strain rate.

**Figure 16 materials-13-04730-f016:**
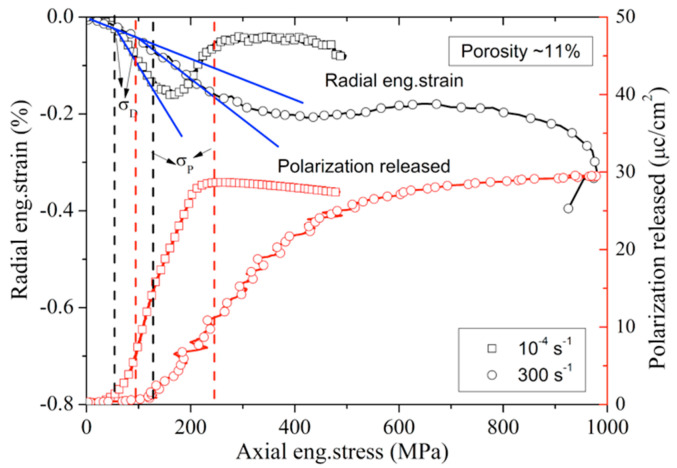
Eng.stress vs. radial eng.strain and eng.stress vs. polarization released curves of porous poled PZT95/5 ceramics at different strain rates.

**Figure 17 materials-13-04730-f017:**
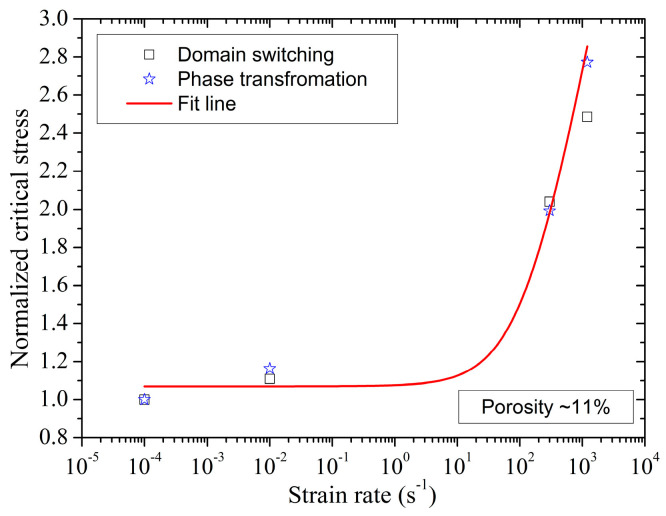
Normalized critical stress of domain switching and phase transformation as a function of strain rate for porosity poled PZT95/5 ceramics.

**Table 1 materials-13-04730-t001:** The value of calculated porosity.

PMMA (wt %)	Density (g/cm^3^)	Porosity (%)
0	7.61	≈5
1	7.13	≈11
2	6.81	≈15
3	6.49	≈18

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
