# Peer review of "Strain Rate and Porosity Effect on Mechanical Characteristics and Depolarization of Porous Poled PZT95/5 Ceramics"

_materials, 2020, doi:10.3390/ma13214730_

Round 1

Reviewer 1 Report

Dear authors,

thank you very much for the intersting work regarding the behavior of porous ceramics under high strain rates.

It is well written and closes a lack in knowledge as you stated.

I have two small comments:

ln 81: pleas add a table for the values of porosity.

I suggest to add a table for the data given in the underline of figure 1 and add some more references to the paper.

Please add more references for the DIC und high strain rates and the used high speed testing setup.

I think the paper is of interest to the reader, due to the closing of the gap in knowledge regarding the strain rates.

Best regards

a Reviewer

Author Response

Dear reviewer 1:

Thank you very much for your review and instructive suggestions with regard to our manuscript “Mechanical response and deformation mechanisms of porous PZT95/5 ceramics under shock-wave compression”. Those comments are very helpful for improving our paper. Some of your questions were answered in the attached file.

Thanks for all the help.

Sincerely Yours,

Zhaoxiu Jiang

Nanjing University of Science and Technology, Nanjing, Jiangsu, China

2020.10.04

Reviewer 2 Report

Please see my comments in the attached file.

Author Response

Dear reviewer 2:

Thank you very much for your review and constructive suggestions with regard to our manuscript “Strain Rate Effect on Mechanical Characteristics and Depolarization of Porous Poled PZT95/5 Ceramics”. Those comments are all valuable and very helpful for revising and improving our paper, as well as the importance guiding significance to our researches. Some of your questions were answered in the attached file.

Thanks for all the help.

Sincerely Yours,

Zhaoxiu Jiang

Nanjing University of Science and Technology, Nanjing, Jiangsu, China

2020.10.04

Round 2

Reviewer 2 Report

Please see my comments in the attached file.

Author Response

Thank you again for reading our manuscript “Strain Rate and Porosity Effect on Mechanical Characteristics and Depolarization of Porous Poled PZT95/5 Ceramics” and reviewing it. The main corrections in the paper and responds are shown in the attached file.
